# COSTAR: Dynamic Safety Constraints Adaptation in Safe Reinforcement Learning

## Abstract

Recent advancements in safe reinforcement learning (safe RL) have focused on developing agents that maximize rewards while satisfying predefined safety constraints. However, the challenge of learning policies capable of generalizing to dynamic safety requirements remains largely unexplored. To this end, we propose a novel **CO**ntrastive **S**afe **TA**sk **R**epresentation (COSTAR) framework for safe RL, designed to enhance the generalization capabilities of existing algorithms to dynamic safety constraints, including variable cost functions and safety thresholds.

In COSTAR, we employ a Safe Task Encoder to extract safety-specific representations from trajectory contexts, effectively distinguishing between various safety constraints with contrastive learning. It is noteworthy that our framework is compatible with existing safe RL algorithms and offers zero-shot adaptation capability to varying safety constraints during deployment. Comprehensive experiments show that our COSTAR framework consistently secures high rewards while adhering to dynamic safety constraints, and demonstrates robust generalization capabilities when faced with out-of-distribution (OOD) tasks.

## 1 Introduction

Reinforcement learning (RL) has demonstrated remarkable success across diverse fields such as video gaming (Silver et al., 2017; Vinyals et al., 2019), finance (Hambly et al., 2023), robotics (Morales et al., 2021) and recommendation systems (Afsar et al., 2022). These achievements highlight RL's robust capability in solving complex sequential decision-making problems and navigating uncertain and intricate environments. In traditional RL settings, the agent is permitted unrestricted exploration of the entire state and action space to maximize the expected total reward. Nevertheless, in real-world scenarios, particularly in safety-critical fields, exploration may be substantially constrained. Examples of such domains include autonomous driving (Wen et al., 2020), robot control (Brunke et al., 2022) and aerospace (Dunlap et al., 2023). In these settings, the agent is required to satisfy certain constraints while maximizing the expected total reward, which is a challenge for traditional reinforcement learning algorithms.

To address this challenge, Safe RL algorithms have been developed. The objective of safe RL is to strike a balance between maximizing the expected total reward and satisfying the constraints throughout the decision-making process. In safe RL problems, constraints are typically quantified numerically as *costs*, similar to *rewards*. The upper bound of the costs is defined as safety threshold, which significantly influences the agent's exploration strategy. A higher safety threshold allows the agent greater latitude to violate constraints and take 'UNSAFE' actions for high-risk but potentially high-reward exploration; Conversely, a lower safety threshold necessitates a more safety-centric approach from the agent. Recent advancements in safe RL can be primarily categorized into primal-dual approaches and primal approaches. Primal-dual approaches (Ray et al., 2019; Stooke et al., 2020; Tessler et al., 2018) employ Lagrangian relaxation techniques where safety constraints are added to the optimization objective with corresponding Lagrange multipliers. Then they simultaneously optimize the policy (primal problem) and adjust the importance of safety constraints (dual problem). Primal approaches (Liu et al., 2020; Xu et al., 2021) embed safety constraints directly within the policy optimization process.

However, current safe RL algorithms are hindered by a key limitation: their policies are trained under static safety constraints, requiring retraining from scratch when these constraints change.

This predominant assumption of static safety constraints overlooks the inherent complexity and variability of real-world environments, where safety requirements (including cost functions and safety thresholds) often alter. For instance, considering a self-driving scenario, and when the car is overspeed, the agent receives a cost. In real world, the speed limit of the road changes depending on the width and traffic volume, therefore the cost function is changing. Recent works (Liu et al., 2023; Khattar et al., 2022; Lin et al., 2023; Yao et al., 2024) attempt to address dynamic safety requirements challenge, yet they are limited to threshold-specific adaptations.

In this paper, we introduce the **CO**ntrastive **S**afe **TA**sk **R**epresentation learning (COSTAR) framework, specifically designed to enhance the adaptability of safe reinforcement learning (safe RL) algorithms to dynamic safety constraints, including time-varying cost functions and dynamic safety thresholds. Drawing inspiration from meta RL(Fakoor et al., 2019; Melo, 2022; Zintgraf et al., 2019; Yuan & Lu, 2022), COSTAR treats different cost functions within the same environment as distinct tasks. We leverage a transformer-based Safe Task Encoder to distill task-specific representations from sequences of transition tuples. To further prioritize safety, we incorporate a Safe Residual Block that accentuates critical safety-related information within the task representation. These representations, rich in safety-relevant features, serve as the input for the actor network, facilitating informed decision-making under variable conditions. The primary goal of the Safe Task Encoder is to maximize the mutual information between these representations and their respective tasks, achieved through a contrastive learning strategy that optimizes InfoNCE (Oord et al., 2018), a lower bound for mutual information. Our framework is compatible with any existing safe RL algorithms, thus enhancing their adaptability to dynamic safety constraints. For demonstration, we choose CRPO(Xu et al., 2021) as our baseline . Experiments conducted in Modified Safety-Gymnasium(Ji et al., 2023a) demonstrate the effectiveness of COSTAR over existing safe RL algorithms in scenarios with dynamic safety constraints.

Our main contributions are summarized as:

- We propose a novel safe task representation learning framework that treats dynamic safety constraints as different tasks, incorporating the idea of meta-learning to enhance the adaptability of Safe RL algorithms to dynamic safety constraints.

- We design an efficient Safe Task Encoder, which employs a contrastive learning approach to extract safe task representations from the exploration trajectory and distinguish between various safety constraints.

- Extensive experiments demonstrate that our COSTAR presents superior performance over existing safe RL algorithms with zero-shot adaption capability to dynamic safety constraints without re-training.

## 2 PRELIMINARIES

### 2.1 MARKOV DECISION PROCESS

A reinforcement learning problem is typically formalized as a Markov Decision Process (MDP), represented by the tuple $M = \langle \mathcal{S}, \mathcal{A}, \mathcal{P}, \mathcal{P}_0, \mathcal{R}, \gamma \rangle$, where $\mathcal{S}$ is the state space; $\mathcal{A}$ is the action space; $\mathcal{P} : \mathcal{S} \times \mathcal{A} \times \mathcal{S} \to [0, 1]$ is the transition dynamics of the environment, with $\mathcal{P}(s'|s, a)$ denoting the probability of transitioning to state $s'$ from previous state $s$ given an action $a$; $\mathcal{P}_0 : \mathcal{S} \to [0, 1]$ is the initial state distribution; $\mathcal{R}(s, a) : \mathcal{S} \times \mathcal{A} \to \mathbb{R}$ is the reward function; $\gamma \in [0, 1)$ is the factor discounting the future reward. The agent has a policy $\pi : \mathcal{S} \to \mathcal{P}(\mathcal{A})$ mapping from the state space to a probability distribution over the actions, with $\pi(a|s)$ denoting the probability of selecting action $a$ in state $s$. Starting from the initial state, at each timestep, the agent observes the current state $s$, selects an action $a$ by policy $\pi$, after which the environment transitions to a new state based on transition dynamics $\mathcal{P}$ and returns a reward $r$. The state distribution at timestep $t$, under policy $\pi$, is denoted as $\mu_\pi^t(s)$. To facilitate the optimization of policy $\pi$, we define the state value function $V_\pi(s)$ and the state-action value function $Q_\pi(s, a)$ as follows:

$$V_\pi(s) = \mathbb{E}_{a_t \sim \pi, s_t \sim \mu_\pi^t(s)} \left[ \sum_{t=0}^{\infty} \gamma^t \mathcal{R}(s_t, a_t) \right] \tag{1}$$

$$Q_\pi(s,a) = R(s,a) + \gamma \mathbb{E}_{s' \sim P(s'|s,a)} \Big[ V_\pi(s) \Big] \tag{2}$$

The objective of the agent is to maximize the expected cumulative reward defined as follows:

$$\mathcal{J}_M^0(\pi) = \mathbb{E}_{s_0 \sim \mathcal{P}_0, a_t \sim \pi, s_t \sim \mu_\pi^t(s)} \left[ \sum_{t=0}^{\infty} \gamma^t \mathcal{R}(s_t, a_t) \right] \tag{3}$$

## 2.2 CONSTRAINED MDP WITH DYNAMIC SAFETY CONSTRAINTS

The safe RL problem is formulated within the framework of a Constrained Markov Decision Process (CMDP), represented by the tuple $M^c = \langle \mathcal{S}, \mathcal{A}, \mathcal{P}, \mathcal{P}_0, \mathcal{R}, \gamma, \mathcal{C}, \mathcal{T} \rangle$. Compared with original MDP, CMDP is augmented with a set of cost functions, $\mathcal{C} = \mathcal{C}_1, \cdots, \mathcal{C}_m$, where each $\mathcal{C}_i(s,a) : \mathcal{S} \times \mathcal{A} \to \mathbb{R}$ maps a given state and action pair to a cost. The costs each have upper bounds defined as safety thresholds $\mathcal{T} : \tau_1, \cdots, \tau_m$. In the CMDP framework, when the agent executes an action $a$, it receives reward $\mathcal{R}$ and costs $\mathcal{C}$. Similar to reward, the expected cumulative cost function with respect to $i$-th cost function $\mathcal{C}_i$ is expressed as:

$$\mathcal{J}_{M^c}^i(\pi) = \mathbb{E}_{s_0 \sim \mathcal{P}_0, a_t \sim \pi, s_t \sim \mu_\pi^t(s)} \left[ \sum_{t=0}^{\infty} \gamma^t \mathcal{C}_i(s_t, a_t) \right] \tag{4}$$

The objective of the agent is to maximize the expected cumulative reward while satisfying safety constraints:

$$\max_\pi \mathcal{J}_{M^c}^0(\pi),$$
$$s.t. \quad \mathcal{J}_{M^c}^i(\pi) \leq \tau_i, \forall i = 1, \cdots, m. \tag{5}$$

In COSTAR, we consider safe RL under dynamic safety constraints by extending the cost functions $\mathcal{C}_i(s,a)$ to time-varying functions $\mathcal{C}_i(s,a,t)$, and by sampling safety thresholds from a predefined distribution. We conceptualize this setup as a meta-RL problem, assuming that the CMDP follows a distribution $p(M^c) : M^c \to [0,1]$, where $M_i^c = \langle \mathcal{S}, \mathcal{A}, \mathcal{P}, \mathcal{P}_0, \mathcal{R}, \gamma, \mathcal{C}, \mathcal{T} \rangle$. While these tasks maintain a consistent CMDP framework, they differ in the cost functions $\mathcal{C}$ and safety threshold $\mathcal{T}$. Accordingly, the task distribution can be expressed as $p(M^c) = p(\mathcal{C}, \mathcal{T})$. During meta-training, the agent interact with a sampled CMDP $M_i^c \sim p(M^c)$ from task distribution and get updated; During meta-testing, the trained policy is applied to a task sampled from the same distribution $p(M^c)$. The objective is to learn a policy that can maximize the expected return while satisfying safety threshold under meta-testing tasks:

$$\max_\pi \mathbb{E}_{M^c \sim p(M^c)} \mathcal{J}_{M^c}^0(\pi),$$
$$s.t. \quad \mathcal{J}_{M^c}^i(\pi) \leq \tau_i, \forall M^c \sim p(M^c), i = 1, \cdots, m. \tag{6}$$

## 2.3 CONTEXT-BASED META LEARNER

Context-based meta learner solves meta-RL problem from the perspective of partially observable MDP. The variation in CMDPs (cost functions $\mathcal{C}$ and safety threshold $\mathcal{T}$) are considered as the unobservable part of the state, with the context-based meta-learner aiming to derive task representations using contextual information. Specifically, context-based meta learner employs task encoder to gather information from history trajectories:

$$z_t = E(\{s_i, a_i, r_i, s_{i+1}, c_i, \tau_i\}_{i=0}^t) \tag{7}$$

where $X = \{s_i, a_i, r_i, s_{i+1}, c_i, \tau_i\}_{i=0}^t$ is the history trajectories; $E$ is the task encoder; $z_t$ is safe task representation to express task information. Subsequently, the policy $\pi(a|s,z)$ is conditioned on the latent task representation $z$, facilitating informed decision-making by inferring speculations of the current CMDP $M^c$. During meta-testing, as the agent interacts with the sampled task, it encodes the collected trajectories into the task representation $z$, which helps the policy to adapt to new tasks.

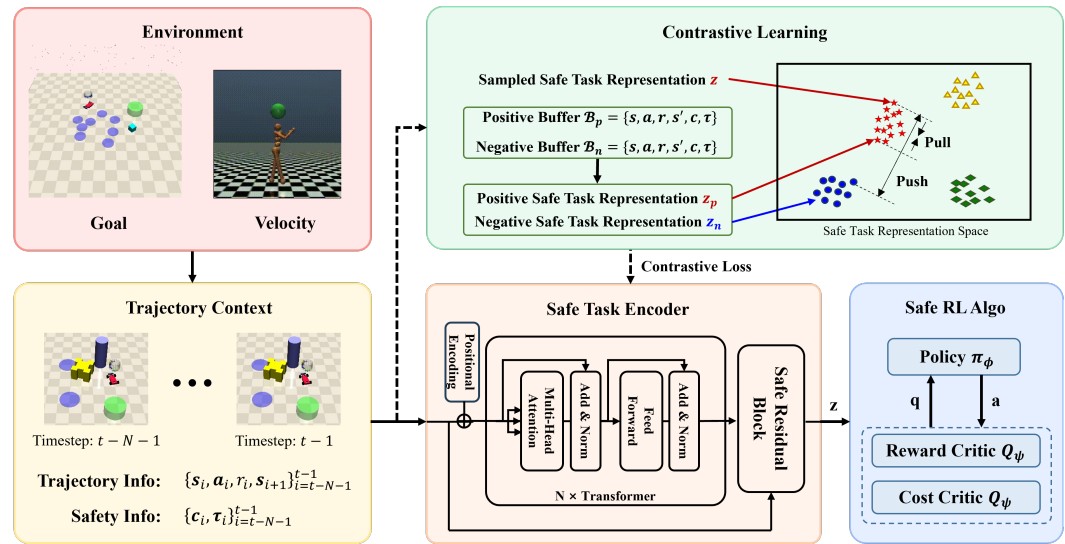

Figure 1: The proposed COSTAR framework. Dashed lines indicate the training process.

## 3 METHOD

To address safe RL problem with dynamic safety constraints, we propose **CO**ntrastive **S**afe **TA**sk **R**epresentation learning (COSTAR), a novel framework for safe task representation learning in safe RL that accommodates tasks with dynamic safety constraints effectively. For dynamic cost functions, we utilize a safe task encoder to extract information from trajectories; For dynamic safety thresholds, a safe residual block is employed to emphasize safety information. The overall structure of COSTAR is illustrated in Figure 1.

### 3.1 SAFE TASK ENCODER

A trajectory context $X = \{s_i, a_i, r_i, s_{i+1}, c_i, \tau_i\}_{i=0}^{t}$ encapsulates the entire process of a CMDP's transitioning from one state to another, containing critical information that characterizes the specific CMDP. Denoting context encoder as $E_\theta$, we define the CMDP $M^c$ as a distribution over encoded latent representation $z = E_\theta(X)$:

$$M^c(z) : \mathcal{Z} \rightarrow [0, 1] \tag{8}$$

where $\mathcal{Z}$ is the context embedding space. In COSTAR, we treat dynamic safety constraints as different CMDPs. Therefore, an important objective of our encoder is to differentiate among CMDPs within the context embedding space $\mathcal{Z}$. For CMDPs that significantly differ, a clear distinction allows the agent to implement CMDP-specific policies; conversely, for similar CMDPs, the agent can effectively leverage shared knowledge. We aim to find an encoder capable of contextualizing trajectory context and distinguishing between CMDPs.

Recently, transformers have demonstrated exceptional performance in the domain of long sequence modeling (Devlin et al., 2018; Dettmers et al., 2022; Wang et al., 2023). Intuitively, given that the trajectory context is presented as a time series, we consider that transformers are well-suited for effectively capturing the underlying relationships between expected cumulative rewards and dynamic safety constraints across various trajectories. In COSTAR, we design a transformer-based encoder to delineate temporal correlations and capture CMDP information within the trajectory context. The proposed safe task encoder comprises a transformer encoder $E_{\theta_1}$ and a safety residual block $E_{\theta_2}$, each characterized by parameters $\theta_1$ and $\theta_2$ respectively. In the transformer encoder $E_{\theta_1}$, two pivotal modules are integral: Multi-Head Self-Attention (MHSA) and Feed-Forward Network (FFN). Each head within the MHSA is designed to focus on different segments of the contextual information, thus

allowing the encoder to comprehensively grasp diverse information dimensions. Subsequently, the FFN introduces non-linearity to the processed features, thereby enhancing the encoder's modeling capabilities. Given the trajectory context $X$, the latent representation $h$ is expressed as

$$h = E_{\theta_1}(X) \tag{9}$$

where $X = \{s_i, a_i, r_i, s_{i+1}, c_i, \tau_i\}_{i=0}^{k}$ and $k$ is the context window size. In safe residual block, a skip connection is established from the safety information $X_s = \{c_i, \tau_i\}_{i=1}^{k}$ to transformer encoder output $h$, thereby generating the safe task representation $z$. This approach aligns with the core advantage of residual blocks, known for their capacity to maintain and amplify essential information across layers, thereby guaranteeing that critical safety elements are fully preserved and enhanced during the model's learning process. The encoded safe task representation $z$ is formulated as:

$$z = h + E_{\theta_2}(X_s) \tag{10}$$

where the architecture of safe residual block $E_{\theta_2}$ is a simple multi-layer perceptron.

## 3.2 SAFE TASK REPRESENTATION LEARNING

To effectively link the safe task representation $z$ with its corresponding sampled Constrained Markov Decision Process (CMDP), we leverage mutual information, a metric quantifying the informational gain about one variable upon observing another. This mutual information serves as a bridge to maximize the association between $z$ and the CMDP $M^c$. Such maximization ensures the encoder not only captures but also preserves critical information, thereby minimizing the uncertainty inherent in the CMDP. More specifically, for a CMDP sampled from CMDP distribution $M^c \sim p(M^c)$, we define the safe task encoder as a probabilistic encoder $z \sim p(z|X)$, where $X = \{s_i, a_i, r_i, s_{i+1}, c_i, \tau_i\}_{i=0}^{k}$ is the trajectory context. During the CMDP process, $X$ is jointly determined by the sampled CMDP $M^c$ and the agent's policy. The learning objective of the safe transition encoder is:

$$\max I(M^c; z) = \mathbb{E}_{M^c, z}\left[ p(M^c, z) \log\left( \frac{p(M^c|z)}{p(M^c)} \right) \right] \tag{11}$$

However, optimizing this objective in practice is infeasible, as we have no access to the joint probability distribution $p(M^c, z)$. Following CPC(Oord et al., 2018), we transform the optimization of mutual information into a binary classification problem as shown in Theorem 3.1.

**Theorem 3.1.** *Consider a set of CMDPs $\mathcal{M}^c = \{M_1^c, M_2^c, \cdots, M_N^c\}$ sampled from $p(M^c)$. Given trajectory context $X = \{s_i, a_i, r_i, s_{i+1}, c_i, \tau_i\}_{i=0}^{k}$ obtained under $M_p^c$, where $M_p^c \in \mathcal{M}^c$. The probability that $M_p^c$ is recognized from $\mathcal{M}^c$ given task representation $z = E_\theta(X)$ can be derived as:*

$$p(M_p^c|\mathcal{M}^c, z) = \frac{\frac{p(M_p^c|z)}{p(M_p^c)}}{\sum_{M_i^c \in \mathcal{M}^c} \frac{p(M_i^c|z)}{p(M_i^c)}} \tag{12}$$

The proof of Theorem 3.1 is given in Appendix A.1.1. By optimizing $p(M_p^c|\mathcal{M}^c, z)$, the safe task encoder will effectively distinguish between positive CMDP $M_p^c$ and the negative CMDP $M_i^c \in \mathcal{M}_n^c = \mathcal{M}^c \backslash \{M_p^c\}$ based on $z$. Following InfoNCE(Oord et al., 2018), we approximate $\frac{p(M^c|z)}{p(M^c)}$ with the exponential of a score function $s(z^*, z)$, which evaluates the similarity between two task representations. The categorical cross-entropy loss of classifying the positive sample correctly is:

$$\mathcal{L}_E = -\mathbb{E}_{M_p^c, z}\left[ \log\left( \frac{\exp(S(z, z_p))}{\exp S(z, z_p) + \sum_{M^c \in \mathcal{M}_n^c} \exp(S(z, z_n))} \right) \right] \tag{13}$$

where $\mathcal{M}^c$ is the set of sampled training CMDPs, $\mathcal{M}_n^c = \mathcal{M}^c \backslash \{M_p^c\}$ is the set of CMDPs other than sampled CMDP $M_p^c$, $z$ and $z_p$ are task representations of trajectory context $X$ and $X_p$, both

derived from the positive CMDP $M_p^c$, respectively. Conversely, the trajectory context $X_n$ is collected under $M_n^c \in \mathcal{M}_n^c$, with corresponding task representation denoted as $z_n$. Following the literature of contrastive learning, we name $(z, z_p)$ a positive pair, and $\{(z, z_n)\}_{M_n^c \in \mathcal{M}_n^c}$ negative pairs. $\mathcal{L}_E$ aims to optimize a categorical cross-entropy loss, enhancing the encoder's ability to distinguish between $(z, z_p)$ and $\{(z, z_n)\}_{M_n^c \in \mathcal{M}_n^c}$, respectively. This optimization improves the encoder's capacity to both extract shared knowledge from trajectory context within the same CMDP and to discern differences in safety constraints among diverse CMDPs.

Meanwhile, $\mathcal{L}_E$ optimizes for a lower bound of mutual information as shown in Theorem 3.2.

**Theorem 3.2.** *Giving a set of CMDP $\mathcal{M}^c = \{M_1^c, M_2^c, \cdots, M_N^c\}$ sampled from $p(M^c)$, $|\mathcal{M}^c| = N$, the mathematical relationship between mutual information $I(M^c; z)$ and loss $\mathcal{L}_E$ is:*

$$\mathcal{L}_E \geq -I(M^c; z) + \log(N) \tag{14}$$

The proof of Theorem 3.2 is given in Appendix A.1.2. While not a prerequisite for training, it is observed that the minimization of $\mathcal{L}_E$ effectively leads to the maximization of a lower bound on mutual information, aligning with our underlying motivation.

---

**Algorithm 1** COSTAR with CRPO(Xu et al., 2021) as safe RL algorithm

---

**Initialize:** policy $\pi$, safety task encoder $E_\theta$, positive buffer $\mathcal{B}$, negative buffer $\mathcal{B}_n$
**Input:** CMDP distribution $\mathcal{M}^c$, training epochs $m$
**for** $epoch = 1$ **to** $m - 1$ **do**
    Sample a positive CMDP $M_p^c \sim \mathcal{M}^c$
    Sample a negative CMDP $M_n^c \sim \mathcal{M}^c \backslash M_p^c$
    Interact with $M_p^c$ and $M_n^c$ and build buffer $\mathcal{B}$ and $\mathcal{B}_n$ respectively
    Run CRPO optimization and update policy $\pi$
    Sample $z$ and $z_p$ from buffer $\mathcal{B}$, and $z_n$ from negative buffer $\mathcal{B}_n$
    Take one-step safe task encoder update towards minimize $\mathcal{L}_E$ according to Eq.13
**end for**

---

### 3.3 Safe Reinforcement Learning Algorithm

Our COSTAR framework is compatible with most safe reinforcement learning algorithms. By simply augmenting the input of actor network and critic networks with safe task representation $z$, COSTAR enables a safe RL algorithm to become adaptable to varying safety constraints. We choose CRPO (Xu et al., 2021) as baseline for demonstration.

For the consistency of notation, we denote state-action critic function in the form of $Q_\pi^i(s, z, a)$, with $i = 0$ indicating the reward critic, and $i = 1, \cdots, p$ indicating the cost critic. Accordingly, the expected total critic function is defined as $J_i(\pi) = \mathbb{E}_{\mathcal{P}_0 \cdot \pi}[Q_\pi^i(s, z, a)]$, where $\mathcal{P}_0$ is the initial state distribution. At each timestep, we check whether there exists an $i \in \{1, \cdots, p\}$ that corresponding $J_i(\pi)$ violates the constraints: $J_i(\pi) < b_i$. If so, the augmented CRPO performs constraint minimization (natural gradient descent on the cost critic) for one of the violated constraints to enforce the safety. If all of the constraints are satisfied, the augmented CRPO performs policy optimization (natural gradient ascent on the reward critic). The algorithmic flow is shown in Algorithm 1.

## 4 Experiment

In this section, we present an empirical validation of our COSTAR, comparing it with the current state-of-the-art methods. We aim to demonstrate: (1)The performance of COSTAR on tasks adaptation in diverse task distributions. (2)The effectiveness of transformer encoder and Safe Residual Block. (3)The ability to adapt to out-of-distribution(OOD) tasks. All experiments are conducted using 5 seeds on a single Nvidia GeForce RTX 3090 GPU.

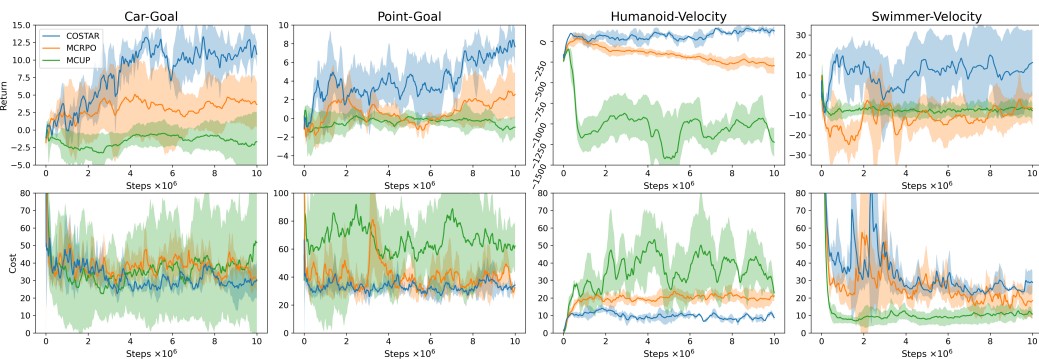

Figure 2: Training results for Modified Safety-Gymnasium benchmarks (Car-Goal, Point-Goal, Humanoid-Velocity and Swimmer-Velocity) with return on the top and cost on the bottom. All subplots represent performance on test tasks over the training steps. The shaded region shows standard deviation across 5 seeds.

## 4.1 EXPERIMENTAL SETTINGS

**Task.** The environments are modified from a publicly available benchmark Safety-Gymnasium(Ji et al., 2023a)[1]. We consider two tasks (Goal and Velocity) and two agents for each task (Point and Car for Goal task, Humanoid and Swimmer for Velocity task). In the Goal task, agents earn rewards for successfully navigating to a designated goal location but incur costs upon entering hazardous areas. In the Velocity task, agent is required to move as quickly as possible while adhering to velocity constraint. These tasks have significant implications in various domains, including robotics, autonomous vehicles, and industrial automation. Based on the original Safety Gymnasium environment, we modified the cost functions and the safety threshold to be dynamic, aligning with our motivation. For comprehensive details on modification and environments, please refer to Appendix A.2. We name the tasks as Car-Goal, Point-Goal, Humanoid-Velocity and Swimmer-Velocity.

**Baselines.** The existing safe RL algorithms only support training at a fixed safety threshold, which is a weakness because in our setting the safety threshold is dynamic and is sampled from a distribution. To make the experiment results comparable, we design a varying safety thresholds extension of traditional safe RL algorithms CRPO(Xu et al., 2021) and CUP(Yang et al., 2022a). During training, their safety thresholds are sampled from the same distribution as COSTAR, which enhance their adaption to varying safety constraints. The modified algorithms are named as MCRPO and MCUP, respectively. We build COSTAR, MCRPO and MCUP on top of the Omnisafe framework (Ji et al., 2023b).

**Metrics.** We compare the methods in terms of episodic reward and episodic cost.

## 4.2 EXPERIMENTAL RESULTS AND ANALYSIS

**Comparison Evaluation.** We compare our COSTAR with MCRPO and MCUP. The evaluation results throughout the training process are shown in Figure 2. COSTAR consistently outperforms the baseline algorithms in most experiments, particularly by achieving higher rewards and maintaining lower costs under dynamic safety constraints. Specifically, in Car-Goal, Point-Goal and Humanoid-Velocity, COSTAR achieves optimal performance, i.e. obtaining the highest rewards while maintaining the lowest costs. In Swimmer-Velocity, COSTAR still achieves the highest reward compared to the baselines, despite incurring higher costs. Notably, COSTAR's costs remain within the established threshold, demonstrating that COSTAR effectively maximizes the use of the safety threshold. Overall, these results underscore COSTAR's effectiveness in adapting to dynamic safety conditions and enhancing performance across varied tasks.

---

[1]The github url of Safety-Gymnasium:https://github.com/PKU-Alignment/safety-gymnasium. We use version 1.2.0 with Apache-2.0 license.

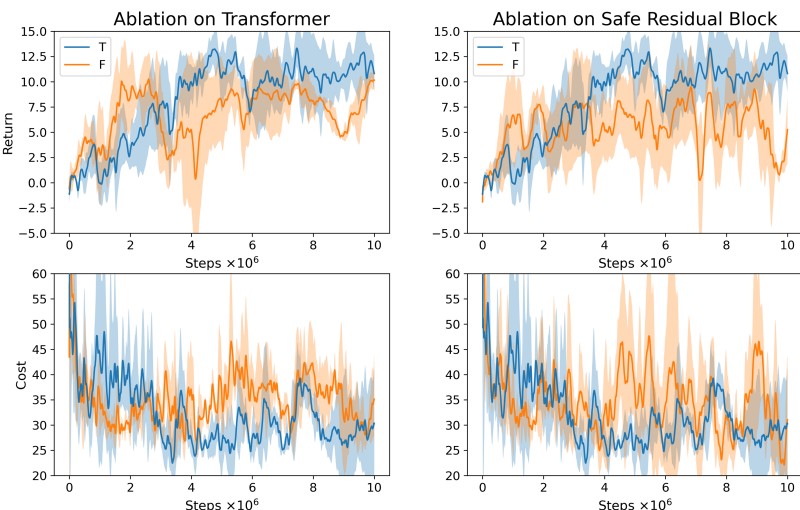

Figure 3: Ablation results on Car-Goal. "T" represents the original COSTAR framework; In the left figure, "F" indicates replacing the transformer with a MLP. In the right figure, "F" indicates removing Safe Residual Block from COSTAR.

**Ablation Evaluation.** To verify the effectiveness of the transformer encoder and the Safe Residual Block, we conducted ablation experiments. For the transformer encoder ablation, we replaced the transformer encoder with a multi-layer perceptron (MLP). For the safe residual block ablation, we removed the safe residual block from the COSTAR framework. The ablation experiment results are shown in Figure 3. We can observe significant performance degradation in both ablation results, demonstrating the significance of both the transformer encoder and the Safe Residual Block. Compared to MLP, transformer encoder is better at capturing complex time-series dependencies and utilizing complex patterns in trajectory. Additionally, removing the Safe Residual Block from COSTAR leads to higher costs and increased instability, highlighting its vital role in reinforcing the integration of safety-related information into the decision-making process. The transformer's sophisticated data processing capabilities and the Safe Residual Block's focus on safety information are both integral to the robust performance of the COSTAR framework.

**Out-of-Distribution Evaluation.** Another critical scenario is how strategies perform in out-of-distribution (OOD) safety constraints. We evaluated the performance of COSTAR with cost functions and safety thresholds that were not encountered during training. The evaluation results for OOD cost functions are shown in Tab 1, where we introduce two new cost functions. Under these conditions, COSTAR exhibits excellent generalizability and stability, achieving maximum rewards without breaching predefined thresholds, and maintaining a minimal standard deviation. This performance underlines COSTAR's robustness and its effectiveness in dynamic environments, consistently delivering

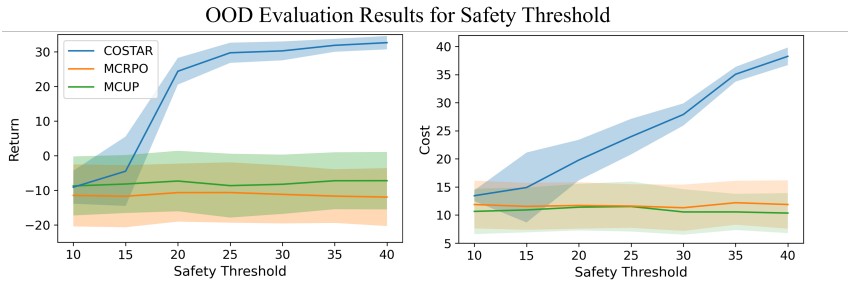

Figure 4: OOD evaluation results for safety threshold in Swimmer-Velocity.

Table 1: OOD Evaluation for Cost Functions in Car-Goal. All values are presented as mean $\pm$ std for 20 episodes. The safety threshold is fixed at 30.

| Cost Function | COSTAR | | MCRPO | | MCUP | |
|---|---|---|---|---|---|---|
| | Return | Cost | Return | Cost | Return | Cost |
| OOD Function A | $12.7 \pm 3.5$ | $28.6 \pm 34.3$ | $7.0 \pm 4.3$ | $36.3 \pm 70.9$ | $-1.2 \pm 0.71$ | $45.6 \pm 117.2$ |
| OOD Function B | $10.1 \pm 4.2$ | $29.4 \pm 35.9$ | $6.8 \pm 4.8$ | $41.9 \pm 44.4$ | $-1.3 \pm 0.73$ | $19.7 \pm 51.0$ |

positive returns even in challenging conditions. The evaluation results for OOD safety thresholds are shown in Fig 4. COSTAR demonstrates a remarkable capability to enhance performance significantly when safety constraints are relaxed, particularly with thresholds above 30, without compromising on safety. Notably, COSTAR not only achieves the highest rewards in evaluations but also adeptly aligns its cost responses to the varying safety thresholds, ensuring an efficient balance between reward optimization and cost management. In contrast, MCRPO and MCUP struggle with adapting to OOD safety constraints, often violating constraints in OOD cost functions and failing to utilize safety thresholds effectively. The details of OOD environments are shown in Appendix A.2.3.

## 5 RELATED WORK

**Safe Reinforcement Learning** endeavors to address reinforcement learning problem through constrained Markov decision processes(CMDP)(Altman, 2021) to mitigate catastrophic exploration behaviors. Currently both primal-dual and primal approaches are commonly employed to accomplish the objective. The primal-dual approaches (Achiam et al., 2017; Tessler et al., 2018; Yang et al., 2020; Stooke et al., 2020) transform constrained optimization problems into a unified formulation through Lagrangian multipliers, optimizing the primal problem (maximizing expected total reward) and the dual problem (embodying safety constraints) simultaneously. Conversely, primal approaches (Liu et al., 2020; Xu et al., 2021; Sootla et al., 2022) focus on the design of objective functions and training process, integrating safety constraints into the optimization process. However, most existing methods are limited by considering fixed safety constraints during training, posing challenges for deployment under dynamic safety constraints. In our work, while our COSTAR utilizes CRPO(Xu et al., 2021) as baseline, our framework can be potentially adapted to most of existing Safe RL algorithms.

**Meta Reinforcement Learning** strives to quickly adapt to new tasks by leveraging training across a distribution of tasks. Optimization-Based approaches(Finn et al., 2017; Gupta et al., 2018; Rothfuss et al., 2018; Al-Shedivat et al., 2017) focus on learning optimal policies across tasks by explicitly modeling the learning process itself, with the goal of finding a meta-policy that can quickly adapt to new tasks with minimal additional learning. Context-Based approaches(Duan et al., 2016; Zintgraf et al., 2019; Fakoor et al., 2019; Rakelly et al., 2019) seek to infer the context of the task through interaction with environment, using this contextual information to guide decision-making. The agent is trained to embed observations from the environment into a context space capturing relevant task information, facilitating policy adaptation to new tasks. In our work, we integrating meta-RL into safe RL, treating dynamic safety constraints as distinct tasks for enhanced adaptability.

**Safe RL with Dynamic Safety Constraints.** Despite significant process in safe RL in recent years, research focusing on the adaptation to varying safety thresholds are scarce(Khattar et al., 2022; Liu et al., 2023). CDT(Liu et al., 2023) employs a decision transformer architecture to allow an agent to dynamically adjust to varying constraint thresholds, though it necessitates additional safety information as input. CWOF(Khattar et al., 2022) employs optimization-based meta-RL techniques to focus on minimizing the upper bounds of task-average optimality gaps and constraint violations. Contrasting with the aforementioned approaches, our COSTAR leverages context-based meta-RL, conditioning the policy on episodic memory generated by the Safe Task Encoder from past experiences. This approach enables us to achieve zero-shot adaptation capability to dynamic safety constraints.

**Contrastive Learning** is a popular method for self-supervised representation learning. It seeks to maximize the similarity between correlated samples and to minimize the similarity between uncorrelated samples for effective data representation learning. (Oord et al., 2018; Grill et al., 2020; Yeh et al., 2022; Wang & Qi, 2022) introduce efficient loss functions in different scenarios. Based

on prior work, contrastive learning has been applied in varieties of domains like computer vision (Wu et al., 2021; Wang et al., 2021; Xie et al., 2021), recommendation (Xie et al., 2022; Yang et al., 2022b; Qiu et al., 2022; Wei et al., 2021), reinforcement learning(Eysenbach et al., 2022; Laskin et al., 2020; Yuan & Lu, 2022). Our goal is to employ contrastive learning for acquiring identifiable task representations for different safety constraints, thereby enhancing the convergence and efficiency of the training process.

# 6  CONCLUSION

The COSTAR framework presents a significant advancement in safe reinforcement learning, offering enhanced adaptability to dynamic safety constraints through a novel transformer-based Safe Task Encoder and a Safe Residual Block. It is noteworthy that COSTAR is compatible with existing safe RL algorithms and possesses zero-shot adaptation capability to varying safety thresholds without re-training. However, the computational requirements may increase because we use transformer to extract safe task representation. Additionally, while COSTAR shows promising results in simulated settings, its performance in real-world applications and under extreme conditions remains to be fully explored. These limitations highlight important areas for future research, particularly in optimizing the framework's efficiency and testing its scalability and robustness in more diverse and challenging environments.

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

# A  APPENDIX

## A.1  PROOF OF THEOREM

### A.1.1  PROOF OF THEOREM 3.1

*Proof.* Consider a set of CMDPs $\mathcal{M}^c = \{M_1^c, M_2^c, \cdots, M_N^c\}$ sampled from $p(M^c)$. Given trajectory context $X = \{s_i, a_i, r_i, s_{i+1}, c_i, \tau_i\}_{i=0}^k$ obtained under $M_p^c$, where $M_p^c \in \mathcal{M}^c$. The probability that $M_p^c$ is recognized from $\mathcal{M}^c$ given task representation $z = E_\theta(X)$ can be derived as:

$$
\begin{aligned}
p(M_p^c | \mathcal{M}^c, z) &= \frac{p(M_p^c|z) \prod_{l \neq p} p(M_l^c)}{\sum_{j=1}^N p(M_j^c|z) \prod_{l \neq p} p(M_j^c)} & (15) \\
&= \frac{\frac{p(M_p^c|z)}{p(M_p^c)}}{\sum_{M_i^c \in \mathcal{M}^c} \frac{p(M_i^c|z)}{p(M_i^c)}} & (16)
\end{aligned}
$$

$\square$

### A.1.2  PROOF OF THEOREM 3.2

*Proof.* Giving a set of CMDP $\mathcal{M}^c = \{M_1^c, M_2^c, \cdots, M_N^c\}$ sampled from $p(M^c)$, $|\mathcal{M}^c| = N$. Let $\mathcal{M}_n^c = \mathcal{M}^c \backslash \{M_p^c\}$ denote CMDPs other than $M_p^c$, and $z$ is obtained under CMDP $M_p^c$. The mathematical relationship between mutual information $I(M_p^c; z)$ and loss $\mathcal{L}_E$ is:

$$
\begin{aligned}
\mathcal{L}_E &= -\mathbb{E}_{M^c} \log \left[ \frac{\frac{p(M_p^c|z)}{p(M_p^c)}}{\frac{p(M_p^c|z)}{p(M_p^c)} + \sum_{M_j^c \in \mathcal{M}_n^c} \frac{p(M_j^c|z)}{p(M_j^c)}} \right] & (17) \\
&= \mathbb{E}_{M^c} \log \left[ 1 + \frac{p(M_p^c)}{p(M_p^c|z)} \sum_{M_j^c \in \mathcal{M}_n^c} \frac{p(M_j^c|z)}{p(M_j^c)} \right] & (18) \\
&\approx \mathbb{E}_{M^c} \log \left[ 1 + \frac{p(M_p^c)}{p(M_p^c|z)} (N-1) \mathbb{E}_{M_j^c} \frac{p(M_j^c|z)}{p(M_j^c)} \right] & (19) \\
&= \mathbb{E}_{M^c} \log \left[ 1 + \frac{p(M_p^c)}{p(M_p^c|z)} (N-1) \right] & (20) \\
&\geq \mathbb{E}_{M^c} \log \left[ \frac{p(M_p^c)}{p(M_p^c|z)} N \right] & (21) \\
&= -I(M_p^c, z) + \log(N) & (22)
\end{aligned}
$$

$\square$

## A.2 ENVIRONMENT DETAILS

### A.2.1 TASK

**Goal.** On a 2D plane of size 3 by 3, there is an agent, a goal and a number of hazards. The agent aims to move towards the goal position while avoiding the hazards. When the agent reaches a goal, the goal's position is randomly reset while preserving the general layout. At each time step, the agent receives a positive reward when approaching the goal, and receives a negative reward when moving away. Each time the Goal is reached, the agent get a positive value of the completed goal reward. If the agent enters hazards area or touches vases, costs will be incurred.

- **Reward Function** The reward function of Goal is defined as:

  (1) reward-distance: At each time step, when the agent is closer to the Goal, it gets a positive value of REWARD, and getting farther will cause a negative REWARD, the formula is expressed as follows

  $$r_t = (D_{last} - D_{now})\beta \tag{23}$$

  Obviously when $D_{last} > D_{now}, r_t > 0$. Where $r_t$ denotes the current time step's reward, $D_{last}$ denotes the distance between the agent and Goal at the previous time step, $D_{now}$ denotes the distance between the agent and Goal at the current time step, and $\beta$ us a discount factor.

  (2) reward-goal: Each time the Goal is reached, get a positive value of the completed goal reward: $R_{goal}$

- **Cost Function** The cost function of Goal is defined as cost-hazards: When the distance of the agent from the center of the hazards $h_{dist} \leq self.size$, the cost is generated: $self.cost \times (self.size - h_{dist})$.

- **Modifications** In the original Safety Gymnasium, the position of hazards are fixed throughout one episode, which means that the cost function is also fixed. To implement dynamic cost functions, we designed two sets of hazards layouts and switched layouts halfway through the episode. Specifically, at timestep 500(the episode length is 1000), the environment changes from layout A to layout B. In this way, the modified cost function is a dynamic time-varying cost function. The coordinates of layout A are: $(0, 0), (1.1, 0), (0.6, \pm0.8), (-0.2, \pm1.0), (-0.8, \pm0.4)$. The coordinates of layout B are : $(-0.5, 0), (0.5, 0), (-1.0, \pm1.0), (0, \pm1.0), (1.0, \pm1.0)$.

  For dynamic safety budgets, the training safety thresholds are sampled from a uniform distribution in the interval $[20, 30]$ per episode.

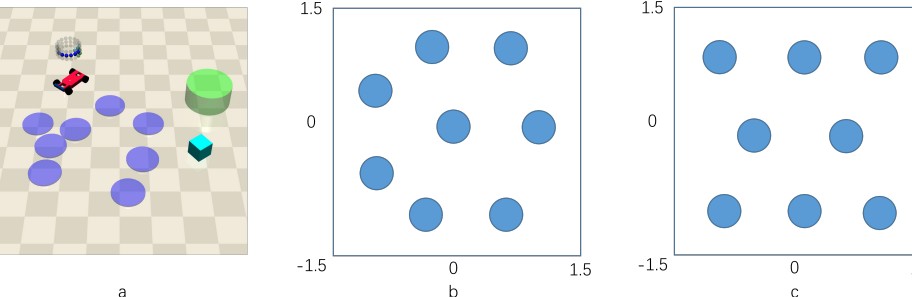

Figure 5: Goal Task. a: The red car is the agent; The green cylinder is the goal; The purple circle is the hazards. b and c are two designed layout A and B respectively.

**Velocity.** The Safe velocity tasks introduce velocity constraints for agents based on the Gymnasium's MuJoCo-v4 series, requiring an agent to move as quickly as possible while adhering to velocity constraint. These tasks have significant implications in various domains, including robotics, autonomous vehicles, and industrial automation.

- **Reward Function** The reward function varies depending on the specific agent, which will be introduced in Appendix A.2.2.

- **Cost Function** The cost function is defined as follow: If velocity of current step exceeds the threshold of velocity, then receive an scalar signal 1, otherwise 0. Formulated as

$$cost = bool(V_{current} \leq V_{threshold})$$

- **Modifications** In the original Safety Gymnasium, the threshold of velocity is fixed through out one episode, which means the cost function is also fixed. To implement dynamic cost functions, we cut the velocity threshold in half at timestep 500(the episode length is 1000). In this way, the modified cost function is a dynamic time-varying cost function.
For dynamic safety budgets, the training safety thresholds are sampled from a uniform distribution in the interval $[20, 30]$ per episode.

### A.2.2 AGENT

**Point** A simple robot constrained to a 2D plane has two actuators, one for rotation and the other for forward/backward movement. This decomposed control scheme makes it particularly easy to control the robot's navigation. It has a small square in front of it, which makes it easier to visually determine the robot's orientation

- **Action Space** (-1.0, 1.0, (2), float64)
- **Observation Space** (-inf, inf, (24,), float64)

**Car** A moderately intricate robot designed for movement in three dimensions features two parallel wheels that can be independently driven, accompanied by a free-rolling rear wheel. In this robot, the coordination of the two drives is essential for both steering and forward/backward movement. Its design bears resemblance to a basic educational robot.

- **Action Space** (-1.0, 1.0, (2), float64)
- **Observation Space** (-inf, inf, (24), float64)

**Humanoid** The 3D bipedal robot is designed to simulate a human. It has a torso (abdomen) with a pair of legs and arms. The legs each consist of three body parts, and the arms 2 body parts (representing the knees and elbows respectively). The goal of the environment is to walk forward as fast as possible without falling over.

- **Action Space** (-0.4, 0.4, (17), float32)
- **Observation Space** (-inf, inf, (376,), float64)
- **Reward Function**
(1)healthy-reward: Every timestep that the humanoid is alive (see section Episode Termination for definition), it gets a reward of fixed value healthy-reward.
(2)forward reward: A reward of walking forward which is measured as forward-reward-weight * (average center of mass before action - average center of mass after action)/dt. dt is the time between actions and is dependent on the frame-skip parameter (default is 5), where the frametime is 0.003 - making the default dt = 5 * 0.003 = 0.015. This reward would be positive if the humanoid walks forward (in positive x-direction).
(3)ctrl-cost: A negative reward for penalising the humanoid if it has too large of a control force. If there are nu actuators/controls, then the control has shape nu x 1. It is measured as ctrl-cost-weight * sum(control$^2$).
(4)contact-cost: A negative reward for penalising the humanoid if the external contact force is too large. It is calculated by clipping contact-cost-weight * sum(external contact force$^2$) to the interval specified by contact-cost-range.
The total reward returned is reward = healthy-reward + forward-reward - ctrl-cost - contact-cost.

**Swimmer** The environment aims to increase the number of independent state and control variables as compared to the classic control environments. The swimmers consist of three or more segments ('links') and one less articulation joints ('rotors') - one rotor joint connecting exactly two links to form a linear chain. The swimmer is suspended in a two dimensional pool and always starts in the same position (subject to some deviation drawn from an uniform distribution), and the goal is to move as fast as possible towards the right by applying torque on the rotors and using the fluids friction.

- **Action Space** (-1.0, 1.0, (2), float32)
- **Observation Space** (-inf, inf, (8,), float64)
- **Reward Function**
  (1)forward-reward: A reward of moving forward which is measured as forward-reward-weight * (x-coordinate before action - x-coordinate after action)/dt. dt is the time between actions and is dependent on the frame-skip parameter (default is 4), where the frametime is 0.01 - making the default dt = 4 * 0.01 = 0.04. This reward would be positive if the swimmer swims right as desired.
  (2)ctrl-cost: A cost for penalising the swimmer if it takes actions that are too large. It is measured as ctrl-cost-weight * sum(action2) where ctrl-cost-weight is a parameter set for the control and has a default value of 1e-4
  The total reward returned is reward = forward-reward - ctrl-cost

### A.2.3 OUT-OF-DISTRIBUTION SETTINGS

**OOD Cost Functions** As mentioned in A.2, in Goal environment, we change the layout from layout-A to layout-B at timestep 500 during training. During OOD evaluations, we designed two new cost functions.

- **OOD Function A** We change the layout from layout-A to layout-B at timestep 200.
- **OOD Function B** We change the layout from layout-B to layout-A at timestep 500.

**OOD Safety Thresholds** The training thresholds are sampled from a uniform distribution in the interval $[20, 30]$. During OOD experiments, we evaluate the performance of trained model under threshold $\{10, 15, 35, 40\}$.

### A.3 EXPERIMENT SETTINGS

In Table 2, we list the important configurations and hyperparameters in training process.

Table 2: Configurations and hyperparameters used in training to produce all the experimental results.

| Configurations | COSTAR | MCRPO | MCUP |
|---|---|---|---|
| Training steps | 1e7 | 1e7 | 1e7 |
| Batch size | 256 | 256 | 256 |
| Gamma | 0.99 | 0.99 | 0.99 |
| Cost gamma | 0.99 | 0.99 | 0.99 |
| Safety threshold | [20, 30] | [20, 30] | [20, 30] |
| Actor type | Gaussian policy | Gaussian policy | Gaussian policy |
| Actor hidden size | [64, 64] | [64, 64] | [64, 64] |
| Actor learning rate | - | - | 0.001 |
| Critic hidden size | [64, 64] | [64, 64] | [64, 64] |
| Critic learning rate | 0.001 | 0.001 | 0.001 |
| Encoder batch size | 64 | - | - |
| Encoder hidden size | 32 | - | - |
| Encoder transformer layers | 3 | - | - |
| Encoder transformer heads | 4 | - | - |
| Encoder learning rate | 0.0003 | - | - |
| Context length | 16 | - | - |

