# OpenReview forum: "COSTAR: Dynamic Safety Constraints Adaptation in Safe Reinforcement Learning"
_ICLR.cc/2025/Conference — ICLR 2025 Conference Withdrawn Submission_

### Official Review · Reviewer_F6jt · 2024-10-27

**Soundness:** 2
**Presentation:** 2
**Contribution:** 2
**Rating:** 3
**Confidence:** 4

**Summary:**

This paper introduces COSTAR (COntrastive Safe TAsk Representation learning), a framework designed to enhance safe RL under dynamic safety constraints. Inspired by meta-RL, COSTAR treats varying cost functions as distinct tasks and uses a transformer-based encoder and a Safe Residual Block to create safety-focused task representations. These representations guide decision-making and improve adaptability to changing conditions. Compatible with existing safe RL methods, COSTAR, demonstrated with CRPO in Modified Safety-Gymnasium, outperforms other algorithms in scenarios with dynamic safety constraints.

**Strengths:**

1. The motivation for addressing dynamic constraints is clearly justified, as a generalizable and adaptable policy is particularly valuable in safety-critical scenarios.
2. The design of the method appears reasonable, introducing two key components that seem well thought out.

**Weaknesses:**

1. The formulation and analysis of COSTAR are restricted to a finite set of cost functions and safety thresholds, limiting its demonstrated flexibility.
2. The evaluation is confined to only two tasks, leaving the broader performance and generalizability of COSTAR less clear.

**Questions:**

1. COSTAR addresses dynamic safety constraints, encompassing both varying cost functions and thresholds. Could the authors clarify if both the cost functions and thresholds are time-varying, or if only the cost functions are intended to vary over time?
2. The abstract mentions that COSTAR demonstrates robust generalization on out-of-distribution (OOD) tasks (lines 23-24). However, during meta-testing, the trained policy is applied to tasks sampled from the same distribution as in meta-training (lines 138-141). Could the authors clarify what is meant by OOD tasks in this context?
3. In Theorems 3.1 and 3.2, why is the sampled set $\mathcal{M}^c$ finite? In general, it is more reasonable to assume this to be infinite, considering the number of cost functions and safety thresholds and their combinations. If the $\mathcal{M}^c$ is indeed finite, how can COSTAR generalize to OOD tasks, i.e., a new $M^c \notin \mathcal{M}^c$ as the learned representation may not capture the new task? What are the assumptions behind evaluation and training tasks?
4. As shown in Algorithm 1, at each iteration, a CMDP is sampled from a distribution $\mathcal{M}^c$. How is this distribution defined? How efficient is COSTAR if |$\mathcal{M}^c$| is large or infinite?
5. According to lines 311-315, COSTAR performs constraint minimization for “one of the violated constraints” to enforce safety. Could the authors clarify how this update rule ensures overall constraint satisfaction, as addressing a single constraint may not satisfy others, particularly if constraints are conflicting with each other [1]? \
[1] https://proceedings.mlr.press/v164/huang22a/huang22a.pdf
6. Regarding the evaluation in Section 4: a) Why are the original environments modified to use fixed obstacle layouts? Randomly sampling the layout at certain time-steps could provide a more rigorous test and a fixed layout could oversimplify the evaluation. b) Could the authors provide more detailed results for Figure 3? It appears that removing certain components of COSTAR does not lead to constraint violations.; c) In Figure 4, COSTAR struggles to maintain safety when the safety threshold is lower than the thresholds used in training (e.g., threshold = 10). Could the authors include additional results for thresholds below the training levels to clarify the method's behavior under stricter safety conditions? d) According to Equation (6), COSTAR should handle multiple constraints simultaneously. Why is it evaluated in environments with only one constraint, which may limit its demonstrated potential? e) The evaluation covers only four environments in total. Could the authors test the method in a wider range of environments with more diverse cost functions and safety thresholds to better showcase COSTAR’s performance?

---

### Official Review · Reviewer_v9WT · 2024-11-02

**Soundness:** 2
**Presentation:** 2
**Contribution:** 2
**Rating:** 5
**Confidence:** 4

**Summary:**

This paper proposes COSTAR, a framework for adaptive safe RL that generalizes to varying safety constraints. COSTAR uses a task encoder to extract safety information from transitions, enabling zero-shot adaptation to new safety requirements. Combined with CRPO, it demonstrates good adaptability without needing fixed constraints.

**Strengths:**

(1) Good motivation: The topic of zero-shot adaptation to various constraints in safe RL is interesting

**Weaknesses:**

(1) Limited evaluation: The proposed method is only compared with two safe RL baselines: the variants from CRPO and CUP. I suggest the authors compare the methods with more up-to-date RL baselines such as CDT [1] and TREBI [2] with adaptive threshold conditions.

(2) Large variance in the experiments: the variance for cost in Table 1 is larger than the mean value. This shows that the proposed method may not be stable.

Reference:

[1] Zuxin Liu, et al. "Constrained decision transformer for offline safe reinforcement learning." International Conference on Machine Learning. PMLR, 2023.

[2] Qian Lin,  et al. "Safe offline reinforcement learning with real-time budget constraints." International Conference on Machine Learning. PMLR, 2023.

**Questions:**

(1) Experiment details: What are the thresholds you used in Figure 2? Can you also visualize the threshold as dash lines in the cost figures?

---

### Official Review · Reviewer_VNtA · 2024-11-04

**Soundness:** 2
**Presentation:** 2
**Contribution:** 1
**Rating:** 3
**Confidence:** 4

**Summary:**

This work targets the problem of CMDP under the dynamic safety constraints setting where the cost function is time-varying and the thresholds in constraints are sampled from a predefined distribution. The authors propose a contrastive learning based approach that learns the representation of task and safety information from trajectories.

**Strengths:**

The setting of CMDP with varying cost functions and thresholds seems new and not explored before.

**Weaknesses:**

The major contribution comes from a contrastive learning formulation for CMDP task representation learning. All the techniques are from existing literature and thus the novelty is very limited.

Although there are some existing results in terms of the mutual information estimation in the manuscript, they are not related to the optimality of CMDP learning.

There is no theoretical analysis on whether the proposed method can learn the optimal feasible policy in CMDP on target test tasks sampled from the same distribution during training.

Question:

In Eq. (6), the threshold is fixed for an episode, while in Eq. (7) and later, it seems that the threshold depends on individual transition as the subscription i here denotes the time step inside an episode. It looks ambiguous as I do not understand what the corresponding CMDP would be if the threshold also depends on the time step. Could the authors please explain this in detail?

**Questions:**

Please see the weakness part.

---

### Official Review · Reviewer_GekV · 2024-11-05

**Soundness:** 2
**Presentation:** 2
**Contribution:** 1
**Rating:** 1
**Confidence:** 5

**Summary:**

This paper proposes a meta learning based safe RL method to make RL agent adapt various safety constraint conditions.

The proposed method utilizes contrastive learning to infer the given context of safe RL, such as cost threshold.

To adapt the variation of safety constraints, the authors introduce a task encoder $E: (s,a,r,s',c,\tau) \to z$ where $\tau$ is the threshold condition.

Then, the proposed policy uses $z$ as an input with state $s$ to satsify the varing safety contexts.

The safe task representation $z$ is trained by contrastive learning and computed with the transformer architecture.

In experiment, the authors conduct the comparison in the part of  safety-gymnasium enviornments.

**Strengths:**

This paper has the following strengths.

1. the proposed architecture is simple to use in practice.
2. The scenario of meta safe RL is quite important to deploy RL agent in the wild.

**Weaknesses:**

The main conern that I have on this paper is that the quality of contribution, presentation, and experiment is not enough to satisfy the standard of this venue.

The detailed comments are as follows.

1. The comparison with existing works are very limited. For example, MESA [1] and LUSR [2] are the very similar methods with slightly different target scenario. MESA is an offline meta safe RL for falut tolerance for few-shot adaptation, and LUSR is an econder based RL adaptation method for generalized state representation.
2. The proposed method seems a simple combination of encoder, constrative learning, and augmented input for policy.
3. One of the main result, figure 4 has no meaningful contribution. The safe velocity environment has the property that the the cost is given when the current velocity is higher than the thresold, and the reward becomes higher when the value of velocity for the foward direction is higher. To show more meaningful result, the authors should have provided the result of the oracle for each single threshold to validate the proposed method succesfully works with adapting the varying safety threshold.
4. The reported environments covers a  small subset of safety-gymnasium enviornments compared to current safe RL papers. It is better to conduct more experiments with unconvered environments to validate the proposed method.


[1] Luo, Michael, et al. "Mesa: Offline meta-rl for safe adaptation and fault tolerance." arXiv preprint arXiv:2112.03575 (2021).

[2] Xing, Jinwei, et al. "Domain adaptation in reinforcement learning via latent unified state representation." Proceedings of the AAAI Conference on Artificial Intelligence. Vol. 35. No. 12. 2021.

**Questions:**

See the above weakness part

---

### Note · Authors · 2024-11-21

I have read and agree with the venue's withdrawal policy on behalf of myself and my co-authors.